# Time Classification Algorithm Based on Windowed-Color Histogram Matching

**Hye-Jin Park, Jung-In Jang and Byung-Gyu Kim *** 

Department of IT Engineering, Sookmyung Women's University, 100 Chungpa-ro 47 gil, Yongsna-gu, Seoul 04310, Korea; hj.park@ivpl.sookmyung.ac.kr (H.-J.P.); ji.jang@ivpl.sookmyung.ac.kr (J.-I.J.)
\* Correspondence: bg.kim@sookmyung.ac.kr; Tel.: +82-2-2077-7293

**Abstract:** A web-based search system recommends and gives results such as customized image or video contents using information such as user interests, search time, and place. Time information extracted from images can be used as a important metadata in the web search system. We present an efficient algorithm to classify time period into day, dawn, and night when the input is a single image with a sky region. We employ the Mask R-CNN to extract a sky region. Based on the extracted sky region, reference color histograms are generated, which can be considered as the ground-truth. To compare the histograms effectively, we design the windowed-color histograms (for RGB bands) to compare each time period from the sky region of the reference data with one of the input images. Also, we use a weighting approach to reflect a more separable feature on the windowed-color histogram. With the proposed windowed-color histogram, we verify about 91% of the recognition accuracy in the test data. Compared with the existing deep neural network models, we verify that the proposed algorithm achieves better performance in the test dataset.

**Keywords:** time classification; sky region; windowed-color histogram; weighting approach



## 1. Introduction

Generally, we design a web-based search system to get accurate answers for the given queries. Meanwhile, the type of search query has diversified in web and mobile environments. In the past, only keywords were used as input query but nowadays it has been extended to sentence, voice, image, and video [1–4]. In addition, by expanding the definition of semantic search, the searching service is developing into a customized service that not only shows objective results but also recommends information that users may like [5]. For example, they recommend and show customized image or video content using information such as user interests, search time, climate, and the place where the user resides.

To use this function, it is essential to use some additional contextual information. Researchers has actively conducted to apply deep learning to extract context information from text, image, video, and voice data. In particular, by using contextual information based on image or video data, it has so far been successfully applied to computer vision tasks such as object detection, semantic segmentation, and image classification [6–12]. This information can describe objects' class, background, as well as some situations or relationship between them [13–15].

There are some useful components in image and video processing. For a long time, texture and color components have had low-level key features for scene description [15]. By using this information, an image descriptor called Contextual Mean Census Transform (CMCT), which combines distribution of local structures and contextual information, was designed to classify scenes. However, it needs to understand a more high-level context, such as the relationship between people or their emotions in the scene. This means that the traditional feature-based approaches cannot support making high-level understanding. Therefore, more high-level abstractive information such as when (time), where (location),

what to do (action), with whom (relationship with other) and so on should be extracted from the image and video data [16–18].

In order to obtain high-level abstracted information, we focus on extracting time information from images. It can be used as important metadata in the search or recommendation system. To extract it from image data, we paid attention to the amount of sunlight because it causes the change of time during the day—dawn, morning, midday, evening, and night. In this aspect, the sky region represents the change of time in outdoor images well and can provide useful information about the environment, especially for the given time. Therefore, by extracting sky area and analyzing its characteristics, it is able to derive meaningful time information during the day.

In this paper, we propose an efficient algorithm to classify time by analyzing the sky region with deep learning and color histogram. The proposed algorithm is trained by Mask R-CNN [19] to extract the sky region from the given image. Based on the extracted region, the reference color histograms are generated, which can be considered as the ground-truth. Time is classified by comparing the reference color histogram with a histogram of the sky region extracted from test dataset. We design the windowed-color histograms (for RGB bands) to compare each time period with the reference and input efficiently. In addition, we propose a weighted histogram representation which compares histograms by assigning large weights according to the degree of color importance.

The main contribution of this study is the followings:

- We focus on the sky area of the image because it has more information on light and discriminating feature;
- The designed weighted histogram comparison can pay attention on more important colors in each class;
- As a result, we simplify the problem and improve the time classification performance better than existing deep learning models.

This study can have a wide range of applications in the real world. If time information is used, a web-based search system can categorize results into dawn, dusk, day, or night. In addition, it can recommend contents based on the related time of queries. We aim to provide more accurate search and recommendation results by using time information of image or video.

The remaining part of this paper is organized as follows: In Section 2, we analyze some existing methods as the related works. Section 3 describes the proposed approach and Section 4 describes the experimental results. Conclusions are presented in Section 5.

## 2. Related Works

Some researches for time classification of the single image have been conducted relying on computer vision algorithms. We categorized into three types: traditional image processing techniques, deep learning-based models, sky detection-based algorithms.

### 2.1. Traditional Image Processing Techniques

Traditional image processing techniques for time classification have employed mainly histograms of images [20–22]. Saha et al. classified daytime and nighttime using thresholding HSV histogram. This method determined the first parameter, representing an amount of red or yellow pixels [20]. In addition, they determined the second parameter of light. This method used brightness and color components, which are important for classifying time. Similarly, Taha et al. used a HSV histogram and discriminated daytime and nighttime scenes employing thresholding H-histogram and V-histogram [21] . In addition, Park et al. proposed a time classification method that used intensity, chromaticity in RGB component, and *k*-means segmentation [22].

However, these methods can only classify binary class. Because the difference between day and night is clear, it is easy to separate. However, it is still challenging to classify day, dawn (dusk), and night. We use a RGB color histogram representing the difference of time period and successfully classify them with high accuracy.

### 2.2. Deep Learning-Based Models

Deep learning-based image analyses have shown a large progress in computer vision [23–31]. Deep models can recognize various features such as low-level as well as high level features. Volokitin et al. and Ibrahim et al. have used deep learning models to get more complex feature and classified time into two or more classes [30,31].

Volokitin et al. first attempted at predicting time classification in outdoor scenes [30]. They employed pooling layers that provide better features than fully connected layers for these tasks. They obtained performance, especially at month and season level. However, the accuracy on the time of day was relatively low.

Ibrahim et al. have proposed a model to extract a visual condition such as dawn/dusk, day, and night for time detection. They trained a model relying on residual learning using the ResNet50 architecture [31]. It gave a little lower accuracy of time classification. We also conducted an experiment employing ResNet152 architecture and fine-tuning our own dataset. The proposed algorithm achieves better performance than ResNet, which has been used in [31].

### 2.3. Sky Detection-Based Algorithms

To estimate the time period of a day, the sky area contains meaningful information due to the change of sun light. Therefore, we noted that sky region extraction is an important module as the pre-processing of time classification. There are some works to detect the sky region in an image [32–36].

Shen et al. used a histogram to extract the sky region [32]. However, there are several problems. First, it takes time to detect the sky region. Second, it often fails to detect the extract sky region. Therefore, we employ Mask R-CNN, a well known semantic segmentation way to resolve these problems [19]. Sharma et al. have made their own dataset using metadata and classified the time segment using deep-learning when an input containing a sky region was given [37].

However, it also has the following problems: First, the metadata is not exact. Second, there is an ambiguity in the categories. Third, the accuracy of classifying the time stamps was not good due to two previous reasons. Therefore, we are going to use only the sky region of the image to classify time variation. We propose an efficient algorithm to classify time by analyzing the color histogram of the sky region.

## 3. Proposed Time Classification Algorithm

Figure 1 shows the overall procedure of the proposed time classification algorithm based on the sky information. In the training step, the dataset including the sky is trained using Mask R-CNN to detect the sky region. A reference color histogram representing the time period (dawn, day, and night) is created based on the sky information. In the evaluation step, the sky region is extracted from test image through the trained Mask R-CNN and a color histogram of the sky area is generated. Finally, the test image is classified into dawn, day, and night in the comparison process between the color histogram of the test image and the reference color histogram.

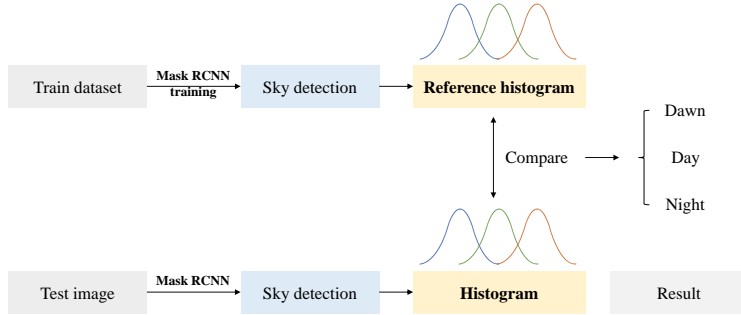

**Figure 1.** The flow chart of the proposed algorithm.

### 3.1. Sky Detection

For sky region detection, the Mask R-CNN [19] is trained by using a dataset consisted of images and annotations of the sky region. The Mask R-CNN is well known model in semantic segmentation task and performs well even in small dataset during transfer learning. The model is initialized using the pre-trained MS COCO weight. Additionally, we freeze the head of Mask R-CNN to extract the generalized feature because the weight of the other layers can be trained to fit the sky detection dataset. Figure 2 shows the results of the detected sky region.

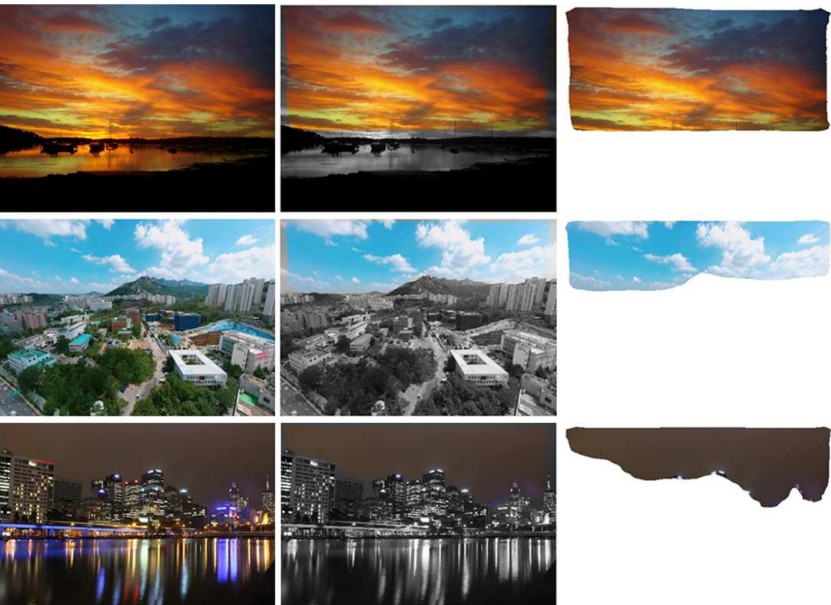

**Figure 2.** The results of the detected sky region using the Mask R-CNN [19].

### 3.2. Time Classification

A reference color histogram representing each time period is generated based on only the sky region extracted from the training dataset. We use four methods to compare the histogram generated from test image with the reference color histogram: Correlation, Intersection, Bhattacharyya, and Chi-square. We find the optimal comparison method and parameters among them. We apply a weighted sum comparison between the color histograms.

#### 3.2.1. Windowed-Color Histogram

With the average of the annotated sky region in train dataset, we construct the reference color histogram of each time period. Figure 3 shows the histograms of dawn, day, and night class. In windowed-color histogram concept, we consider some dominant color parts more when compared with each other. As shown in Figure 4, we take a limited range (limited bandwidth) of the histogram centered on the average value of each color band.

The windowed-color histogram of input image is computed with the pre-defined bandwidth. We compute the matching score with the windowed-color histograms of the references to classify the input image into each time period class, as shown in Equation (1).

$$d_i(H_1^i, H_2^i) = F(H_{1\{-\frac{BW}{2}, \frac{BW}{2}\}}^i, H_{2\{-\frac{BW}{2}, \frac{BW}{2}\}}^i), \tag{1}$$

where $F(\cdot)$ is one of the histogram comparing methods, $BW$ is a pre-defined range centered on the average value of each color, and $i$ is an index of {blue, green, and red} bands.

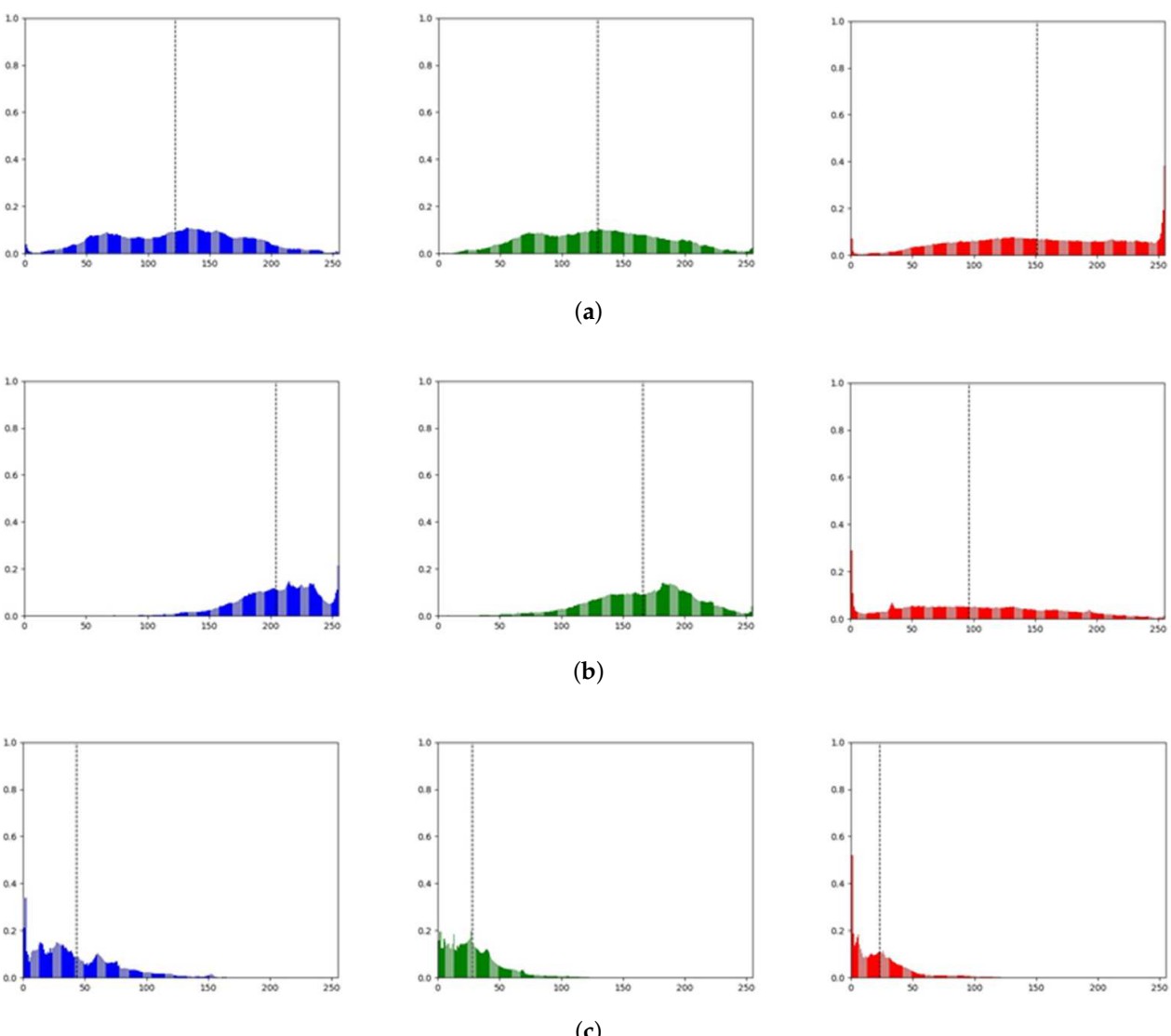

**Figure 3.** Reference color histograms (B, G, R bands) of each time period. (**a**) Reference color histogram of "dawn" or "dusk" time periods. (**b**) Reference color histogram of "day" time period. (**c**) Reference color histogram of "night" time period.

### 3.2.2. Histogram Comparison Methods

- Correlation

Correlation is a statistical technique that shows the relationship between two variables, whether it is linear or non-linear and how strongly they are related. The result of a correlation is called the correlation coefficient and the equation of it is:

$$d(H_1, H_2) = \frac{\sum_I (H_1(I) - \overline{H}_1)(H_2(I) - \overline{H}_2)}{\sqrt{\sum_I (H_1(I) - \overline{H}_1)^2 \sum_I (H_2(I) - \overline{H}_2)^2}}, \tag{2}$$

where $I$ is an index of histogram bins and $H_1(I)$ and $H_2(I)$ are the histograms. In addition $\overline{H}_k$ is the average value of intensity. The denominator of this equation stands for the multiple of the standard deviation of $H_1(I)$ and $H_2(I)$. The numerator stands for the covariance of $H_1(I)$ and $H_2(I)$, so it means the change between $H_1(I)$ and $H_2(I)$. The possible value of the correlation lies between $-1$ and $1$. If two histograms are closely related, $d(H_1, H_2)$ is closer to $+1$ or $-1$. If not related, then $d(H_1, H_2)$ is closer to $0$.

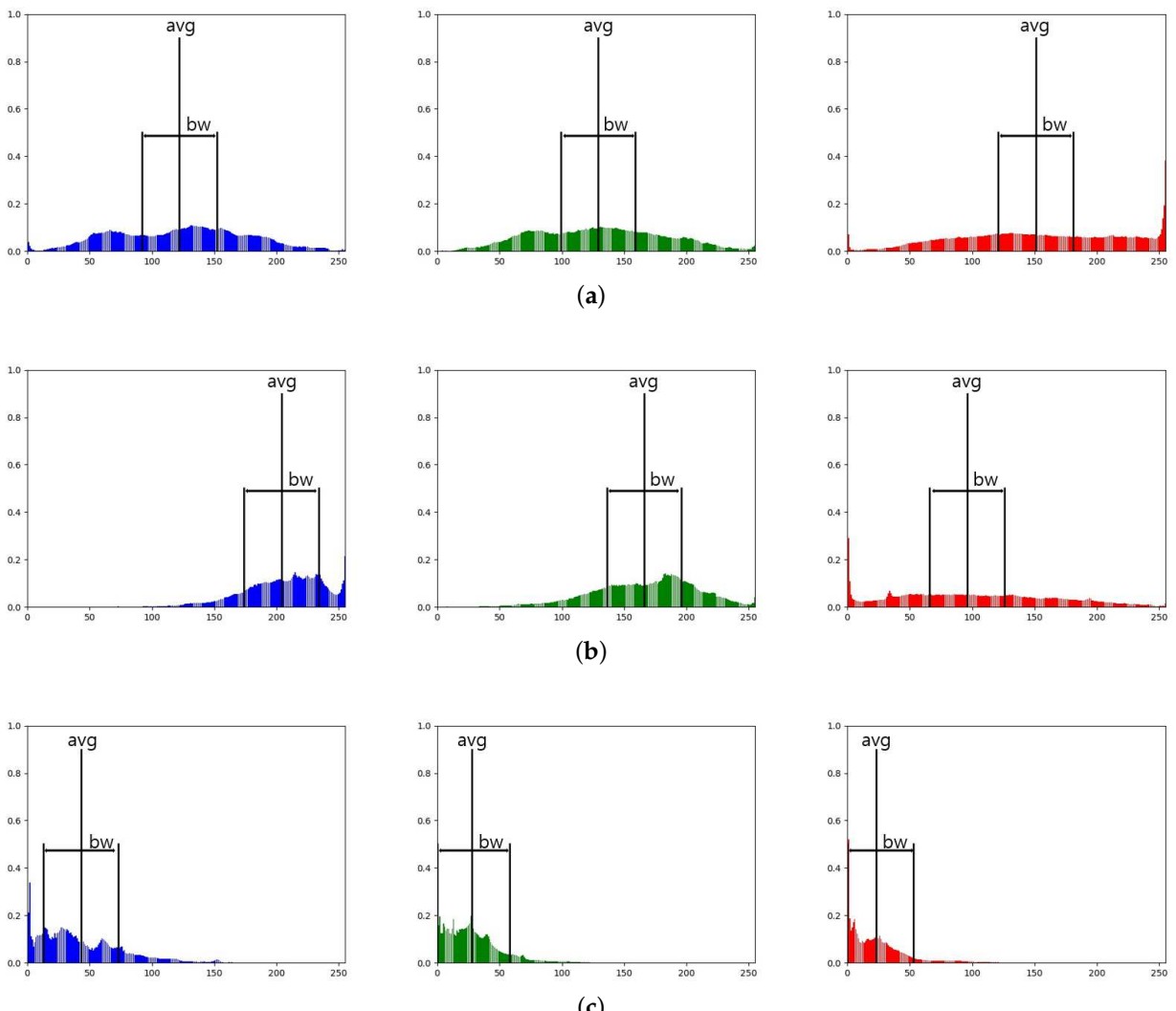

**Figure 4.** The windowed-based color histograms of the generated reference color histograms. (**a**) "dawn" or "dust" case. (**b**) "day" case. (**c**) "night" case.

- Intersection

  Intersection calculates the similarity of two probability distributions or histograms [38]. The equation is:

$$d(H_1, H_2) = \sum_I min(H_1(I), H_2(I)),\tag{3}$$

  where $I$ is an index of histogram bins and $H_1(I)$ and $H_2(I)$ are the histograms. The possible value of the intersection lies between 0 and 1. If two histograms are closely related, $d(H_1, H_2)$ becomes a larger number. If not related, then $d(H_1, H_2)$ is closer to 0, which means there is no overlap.

- Bhattacharyya

  Bhattacharyya distance measures the similarity of two probability distributions or histograms [39]. The Bhattacharyya coefficient is a measure of the amount how two probability distributions or histograms are overlapped. The equation of Bhattacharyya coefficient is:

$$BC(H_1, H_2) = \sum_I \sqrt{H_1(I) \cdot H_2(I)},\tag{4}$$

where $I$ is an index of histogram bins and $H_1(I)$ and $H_2(I)$ are the histograms. As you can see, the Bhattacharyya coefficient stands for the sum of discrete probability distribution, which is overlapped between two probability distributions or histograms and the range is [0,1]. If two histograms are closely related, $d(H_1, H_2)$ is closer to 1. $d(H_1, H_2)$ will be 0, if there is no overlap at all. In fact, Hellinger distance can be defined as:

$$d(H_1, H_2) = \sqrt{1 - BC(H_1, H_2)}. \tag{5}$$

According to the Bhattacharyya coefficient, if two histograms are closely related, $d(H_1, H_2)$ is closer to 0. If not related, then $d(H_1, H_2)$ is closer to 1.

- Chi-Square

Chi-square distance is one of the distance measures that can be used to find dissimilarity between two histograms [40]. The expression can be defined as:

$$d(H_1, H_2) = \sum_I \frac{(H_1(I) - H_2(I))^2}{H_1(I)}, \tag{6}$$

where $I$ is an index of histogram bins and $H_1(I)$ and $H_2(I)$ are the histograms. It means the difference against the denominator histograms. If two histograms are closely related, $d(H_1, H_2)$ becomes 0, because $H_1(I) - H_2(I)$ becomes 0. If not related, then $d(H_1, H_2)$ becomes a larger number.

To summarize, Intersection and Bhattacharyya methods are techniques used to measure the similarity of two histograms by calculating the overlap area. Therefore, these methods are effective for comparing histograms that have different color features.

### 3.2.3. Weighted Histogram Comparison

Brightness of the acquired image depends on the color band during the day. We observe that some specific color components are important to separate the time period. For example, blue and green components can give good separability between "dawn" and "day" periods. However, a red color is not a good indicator to verify them, as shown in Figure 3. Therefore, we design a weighted matching scheme between the reference color histogram and the input image histogram as the following:

$$d(H_1, H_2) = \sum_i w_i d_i(H_1^i, H_2^i) \tag{7}$$

where $d_i$ is defined as the $i$-th band difference in the above, $w_i$ is a tuned weighting factor for each color band. We set this weighting factor $w_i$ through the experiments.

With the above weighted histogram mechanism, we use a hierarchical classification structure: (1) First, the input image is classified into "day (dawn)", and "night" classes based on the windowed-color histogram and weighting factor, (2) After that, the input image, which is in the "day (dawn)" class, is separated into "day" and "dawn" classes based on the refined windowed-color histogram and weighting factor.

## 4. Experimental Results

### 4.1. Experimental Environment

The proposed algorithm has been implemented in Python language. We used i7 CPU with GeForce GTX 1070 GPU (8 G Byte) and 16 GB RAM. Deep learning frameworks for extracting sky area are Tensorflow 1.14.0 and Keras 2.1.0. For training the Mask R-CNN, we used Stochastic gradient descent (SGD) as optimizer and set the same parameters (learning rate = 0.001 and 30 epochs) in [38].

*4.2. Dataset*

The dataset for time classification used a total of 450 images and divided them to 150 images for the training dataset, 300 images for the validation dataset, and 300 images for the test dataset. Images are collected by web crawling and various time scenes are captured from movies. The train and validation dataset are used to extract the sky region using Mask R-CNN. They consist of images, annotations of sky regions, and are classified by time variance: dusk or dawn, day, night. The validation dataset is used only for sky detection using the Mask R-CNN. After training the Mask R-CNN model, the reference histogram of each time class is created to compare the color and brightness of the sky region. Figure 5 shows a part of our dataset for each time period.

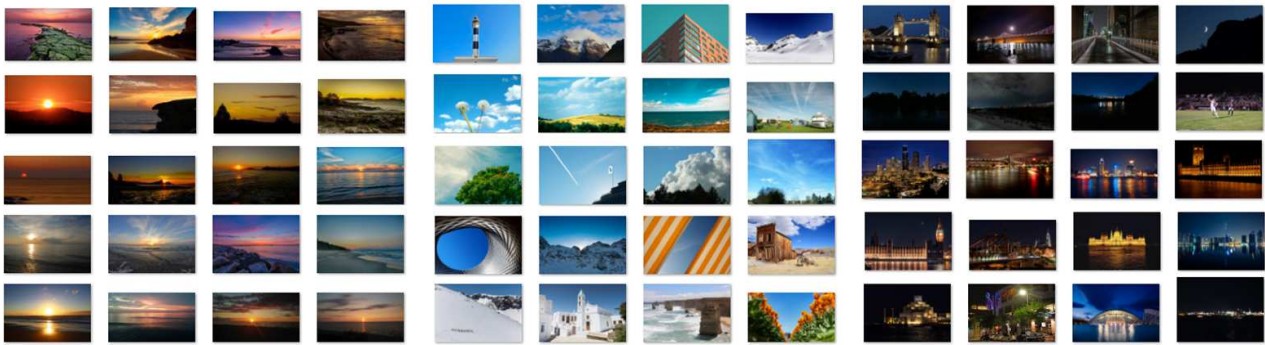

**Figure 5.** Example of each time period from the dataset.

*4.3. Results and Discussion*

4.3.1. Sky Detection

With the validation dataset, we compare the detection accuracy between using Shen's method [32] and the Mask R-CNN method. The detection accuracy is shown in Table 1. The Mask R-CNN achieved 91.67% accuracy. It has much better performance than Shen's method. From this result, we can see the the Mask R-CNN method is very powerful to separate the desired object from the natural background.

**Table 1.** The accuracy comparison between Shen's method [32] and the Mask R-CNN method for sky detection.

| Methods | Accuracy (%) |
|---|---|
| Shen's method [32] | 60.33 |
| Mask R-CNN [19] | 91.67 |

4.3.2. Time Classification

We first classify "night" and "day" using a hierarchical classification structure to simplify the problem. Intersection and Battacharya histogram comparison methods are appropriate to classify images with distinct features. Then, the bandwidth ($BW$) is changed according to the average of each channel of the reference color histogram. As a result of the experiment, we find that the $BW = 120$ and intersection histogram comparison method perform best when classifying images into "night" and "day (dawn)". As mentioned previously, the difference in contribution of each color component (BGR) should be considered.

Table 2 shows the result of each color contribution experiment calculated by the intersection measure. According to the intersection result, green is more important than other colors. Therefore, we calculate the accuracy using the weighted histograms comparison by assigning the largest weight to green. In Table 3, when the weight ratio of blue, green, and red was $\{0 : 6 : 4\}$ and the time classification accuracy was 98.33%, we achieved the best results.

**Table 2.** Result of classifying "night" and "day (dawn)" time periods using intersection with bandwidth = 120.

| [B:G:R] | Intersection (%) |
|---------|------------------|
| 1:0:0 | 92.33 |
| 0:1:0 | 97.33 |
| 0:0:1 | 93.67 |
| 1:1:0 | 96.67 |
| 1:0:1 | 96.67 |
| 0:1:1 | 96.67 |

**Table 3.** Result of classifying "night" and "day (dawn)" time periods using intersection with the weighting approach.

| Weight | Accuracy (%) |
|--------|--------------|
| 0:9:1 | 97.33 |
| 0:8:2 | 97.33 |
| 0:7:3 | 93.00 |
| 0:6:4 | 98.33 |
| 1:8:1 | 97.00 |
| 1:7:2 | 97.00 |
| 1:6:3 | 97.67 |
| 2:7:1 | 97.00 |
| 2:6:2 | 97.00 |

After classifying "night" time, the remaining dataset is classified as "dawn" or "day". Similar to the "night" time classification, the bandwidth was adjusted using a histogram comparison method. As a result, the correlation method performed better than the other methods, including intersection, and it was effective to consider all sections of *BW*.

In Table 4, as a result of calculating the importance of each histogram color in the correlation, the blue color is important to distinguish between day and dawn. It means that the blue color has good separable feature to classify the sky region into "dawn" or "dusk" and "day" time periods. On the other hand, the red color affects less than the blue color. The green color band almost does not affect this classification. Therefore, we assign the largest weight to the blue color band and an accuracy of 91% was achieved when the blue, red, green were weighted by $w_i = \{6 : 1 : 3\}$, respectively as shown in Table 5.

**Table 4.** Result of classifying "dawn" or "dusk" and "day" time periods using the correlation method with all range of the bandwidth.

| [B:G:R] | Correlation (%) |
|---------|-----------------|
| 1:0:0 | 89.67 |
| 0:1:0 | 79.67 |
| 0:0:1 | 48.00 |
| 1:1:0 | 87.00 |
| 1:0:1 | 89.67 |
| 0:1:1 | 78.33 |

Table 6 shows the performance comparison between the proposed method and the existing deep learning classification models. We employed ResNet [6], DenseNet [25], XceptionNet [26], InceptionResNet [27], NasNet [28], and EfficientNet [29]. The same train and validation dataset were used, and the classification performance was inferred from the test dataset. The deep learning models obtained the result with transfer learning using the pre-trained ImageNet weights. Among them, ResNet152 achieved the highest accuracy of 87.67%. The proposed algorithm achieved 91% of accuracy as shown bold face. This means

that the proposed scheme is superior to the existing deep learning models for time period classification even though three classes (day, dawn, and night) exist.

**Table 5.** Result of classifying "dawn" or "dusk" and "day" time periods using the correlation method with the weighting approach.

| Weight | Accuracy (%) |
|--------|--------------|
| 9:1:0 | 90.33 |
| 9:0:1 | 89.33 |
| 8:2:0 | 89.00 |
| 8:1:1 | 90.33 |
| 8:0:2 | 90.67 |
| 7:0:3 | 90.33 |
| 7:1:2 | 90.33 |
| 7:2:1 | 89.67 |
| 7:3:0 | 87.67 |
| 6:0:4 | 89.67 |
| 6:1:3 | 91.00 |
| 6:2:2 | 90.67 |

The existing models have high complexity because they train a whole area of images and find optimal feature. The usual dataset includes various place, season, objects, and backgrounds. Therefore, it needs to focus on the discriminative feature of time information. We simplify this problem through three approaches: sky detection, weighted histogram comparison, and hierarchical classification structure. As a result, we have better performance than other models. However, there are a few things to that need to be improved. If various data can be collected, we can generate a more representative reference histogram for robust classification. In addition, future studies will be needed for more detailed time classification such as dawn, morning, noon, afternoon, evening, and deep night.

**Table 6.** Performance comparison of the proposed algorithm with the existing deep learning classification models.

| Compared Methods | Accuracy (%) |
|------------------|--------------|
| **Proposed algorithm** | **91.00** |
| ResNet152 [6] | 87.67 |
| ResNet101 [6] | 86.00 |
| NasNet [28] | 82.00 |
| EfficientNetB4 [29] | 77.00 |
| EfficientNetB3 [29] | 73.67 |
| EfficientNetB2 [29] | 75.67 |
| EfficientNetB1 [29] | 72.33 |
| EfficientNetB0 [29] | 69.67 |
| InceptionResNet [27] | 74.00 |
| XceptionNet [26] | 73.67 |
| DenseNet201 [25] | 69.67 |
| DenseNet169 [25] | 69.33 |

## 5. Conclusions

This paper has proposed an efficient time-classification algorithm using the sky region detection approach based on a single image. The sky region of an image contains a change in brightness over time. For sky detection, we trained the Mask R-CNN model and made reference color histograms of all time periods (day, dawn, and night). We employed two measures for the proposed windowed-color histogram. Additionally, we designed the weighting factor of each channel to reflect the characteristics of each color and made more efficient classifications. Based on the proposed algorithm, we have achieved 91% of the verification accuracy with our hierarchical classification structure.

**Author Contributions:** Conceptualization, B.-G.K.; methodology, B.-G.K. and H.-J.P.; software, H.-J.P.; validation, J.-I.J. and H.-J.P.; formal analysis, B.-G.K. and H.-J.P.; investigation, J.-I.J. and H.-J.P.; writing—original draft preparation, J.-I.J. and H.-J.P.; writing—review and editing, B.-G.K.; supervision, B.-G.K.; project administration, B.-G.K. All authors have read and agreed to the published version of the manuscript.

**Funding:** This research received funding from the Ministry of Culture, Sports and Tourism (MCST) and from the Korea Copyright Commission in 2021.

**Institutional Review Board Statement:** Not applicable.

**Informed Consent Statement:** Not applicable.

**Data Availability Statement:** Not applicable.

**Acknowledgments:** This research project was supported by Ministry of Culture, Sports, and Tourism (MCST) and from the Korea Copyright Commission in 2021.

**Conflicts of Interest:** The author declares no conflict of interest.

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
