# Peer review of "Time Classification Algorithm Based on Windowed-Color Histogram Matching"

_applsci, doi:10.3390/app112411997_

Round 1

Reviewer 1 Report

Authors study indeed an interesting topic, which could have a wide range of applications in the real world. While the proposed research is well motivated, author can enhance the submission from the following crucial aspects,

The paper needs to give clear statement and introduction to demonstrate its technical 
significance and merits, compared to existing approaches in the area. Thus,
it will be great to add more description in section 1 to highlight core research motivation 
- why the problem addressed in the study is so important? and how the proposed technique 
can benefit or be used in real applications.

In general, journal publication should give very detailed introduction of the related studies published in recent
years. However, the section 2 of the article is not comprehensive enough to provide a good coverage. Most troublesome is
the significant gap in the coverage of the state-of-the-art. A few important literature on deep learning based image analysis is not cited,

BBAS: Towards Large Scale Effective Ensemble Adversarial Attacks against Deep Neural Network Learning

Adaptive and Robust Partition Learning for Person Retrieval with Policy Gradient
.... Many many....

While motivation is strong and the basic techniques developed is interesting, fundamental approaches are
built up with existing algorithm or framework. Good to see more analysis and break-down about core design of the proposed
framework .

Author Response

Thank you for your valuable comments. I have attached a file for replies for comments. Could you refer to the attachment, plz?

Reviewer 2 Report

This work is appropriate for this call, but the manuscript must be enhanced:

  • More references must be cited in the related work section.
  • There are several English mistakes. The whole paper must be checked before the new submission.
  • sufficient background and all relevant references can be improved.
  • The aim of the paper should be clearly indicated.
  • The methods and results described can be improved.

Author Response

Thank you for your valuable comments. I have attached a file for replies for comments. Could you refer to the attachment, plz?

Prof. Kim
